# Effect of Fractal Topology on the Resistivity Response of Thin Film Sensors

**DOI:** 10.3390/s23052409

**Published:** 2023-02-22

**Authors:** Gregory Kopnov, Sudhansu Sekhar Das, Alexander Gerber

**Affiliations:** 1School of Physics and Astronomy, Tel Aviv University, Tel Aviv 69978, Israel; 2School of Physical Science, National Institute of Science Education and Research, Bhubaneswar 752050, India

**Keywords:** thin film sensors, percolation, palladium, hydrogen sensor

## Abstract

We discuss the effect of topological inhomogeneity of very thin metallic conductometric sensors on their response to external stimuli, such as pressure, intercalation, or gas absorption, that modify the material’s bulk conductivity. The classical percolation model was extended to the case in which several independent scattering mechanisms contribute to resistivity. The magnitude of each scattering term was predicted to grow with the total resistivity and diverge at the percolation threshold. We tested the model experimentally using thin films of hydrogenated palladium and CoPd alloys where absorbed hydrogen atoms occupying the interstitial lattice sites enhance the electron scattering. The hydrogen scattering resistivity was found to grow linearly with the total resistivity in the fractal topology range in agreement with the model. Enhancement of the absolute magnitude of the resistivity response in the fractal range thin film sensors can be particularly useful when the respective bulk material response is too small for reliable detection.

## 1. Introduction

Thin films offer significant advantages for sensing applications compared with their bulk alternatives in terms of cost, mechanical stability, and the surface to volume ratio [1,2,3,4,5,6,7]. However, in the very thin limit, the material is not homogeneous covering only a fraction of the surface, and the question is how it affects the sensor’s response. The Volmer–Weber growth mode [8] is common in the deposition of metals on insulating substrates [9]. Films start to grow as statistically distributed and isolated equiaxed islands. During further deposition, these islands increase in size, while the intergranular distances decrease until the islands start to coalesce to form larger but still isolated clusters. By reaching a critical radius, the equiaxed islands change their shape to elongated, which with the further supply of matter interconnect and form a network structure. Reaching the percolation connectivity threshold, a supercluster occurs, spanning the whole substrate. This infinite cluster finally densifies, leading to the formation of a continuous metal film. Thus, the morphology evolves by a gradual transition from isolated equiaxed islands to elongated islands to multiply connected non-percolating islands to a percolating metal film to the filling of the holes [10]. While the early growth is dominated by parameters that are not universal, such as interactions between the material and the substrate, the substrate temperature, and surface tension, it was found that large structures observed near continuity do obey certain universal scaling relations identical to those predicted by the percolation models [11,12].

In the percolation models, the surface fraction x of the system, covered by the film, controls the physical properties. For $x<x_{c}$, only local clusters are present; for $x>x_{c}$ spanning clusters linking the sample borders cover the surface. At the percolation threshold $x_{c}$, the surface coverage results in the first spanning cluster. The percolation transition is a phase transition of second order, as the physical and the geometrical properties of the percolating system diverge or converge continuously at $x_{c}$ with power laws of the quantity $\left(x-x_{c}\right)$. Another basic property of percolating systems is their self-similarity on length scales <ξ, where ξ is the correlation length. ξ diverges at $x_{c}$ but is finite below and above $x_{c}$: $\xi \propto {\left|\;x-x_{c}\right|}^{-\nu }$. Below the continuity (percolation) threshold, the finite clusters present a broad size distribution. For cluster sizes L such that L<ξ, the number $n_{s}$ of clusters of area *s* varies as $n_{s}\propto s^{-\tau }$, where τ is an exponent close to 2. Above the threshold, the infinite cluster has an anomalous scale-dependent density. The mass is not proportional to the area $L^{d}$, where *d* is the usual dimensionality (here *d* = 2), but varies as $L^{D}$, where D is an anomalous, or fractal, dimensionality [13]. This behavior is a straightforward result of the scaling theory of percolation and the assumption that there is only one relevant length scale ξ in the problem, i.e., on scales smaller than ξ, there is no observable characteristic length scale, and the object must be self-similar. Morphology of numerous ultra-thin films at the early stages of growth follow these guidelines, and are characterized as fractal objects [11,12,13,14,15,16,17]. This opens up the possibility of quantitative characterization of these structures and the use of percolation models to predict their transport properties.

An important prediction of the percolation theory is the electrical conductance σ in the vicinity of the percolation threshold: $\sigma \propto {\left(x-x_{c}\right)}^{m}$, with the exponent *m* = 1.3 in two-dimensional (2D) and *m* = 2 in three-dimensional (3D) systems. The critical surface coverage $x_{c}$ can vary significantly, depending on the material, substrate, and conditions of the deposition. For example, $x_{c}$ of Au films is close to 50%, as predicted by percolation theory for a random 2D continuum. On the other hand, indium tends to agglomerate due to its low melting point, and its $x_{c}$ is above 90%. Nevertheless, the critical behavior of the conductivity near the percolation threshold is the same in both materials [18]. This supports the assumption of universality of the critical indices: while the value of $x_{c}$ depends on the short-range correlations in the system and may vary a lot, the value of the critical indices does not.

The issue we address here is how the topological inhomogeneity of very thin metallic conductometric sensors affects their resistivity response to external stimuli, such as pressure, intercalation, gas absorption, or any other mechanism that modifies bulk conductivity. We developed a theoretical model of the response based on the percolation theory. The experimental model system is hydrogenated palladium. Pure palladium (Pd) is a transition metal whose electronic properties are determined mainly by an unfilled d band and, to a good approximation, one can describe the band structure of Pd as a rather narrow 4d band overlapped by a larger 5s band [19]. Pd can absorb and release large amounts of hydrogen at ambient temperature and pressure, which makes it the material of choice in multiple applications of the future hydrogen economy [20]. Absorption of hydrogen leads to a strong lattice expansion [21,22], formation of a so-called H band below the Fermi level, and filling of the d holes by H atoms [23,24]. The electrical resistivity in pure Pd is dominated by the usual scattering of s-electrons by impurities and lattice vibrations. The resistivity of the hydrogenated phase is higher as the H atoms, which occupy the octahedral sites of fcc lattice in a random way, act as scattering centers for the conducting electrons. In bulk Pd, the hydrogen scattering resistivity $\Delta \rho_{H}$ is expected to increase with hydrogen content at a rate $\Delta \rho_{H}/\Delta c_{H}\approx$ 45 µΩcm, where $c_{H}$ is the atomic hydrogen concentration $c_{H}=H/Pd$ [25]. An open question is the magnitude of the H scattering in heterogeneous ultra-thin films. The subject has practical importance since the conductance response of hydrogenated films is frequently used to determine the degree of hydrogen absorption and the phase diagrams of thin film metal–hydrogen systems.

## 2. The Model

As discussed in the introduction section, it was realized long ago that the geometrical structure and properties of multiple heterogeneous systems can be described by fractal topology models. In particular, the electrical and thermal conduction in very thin films and random and granular mixtures of elements with different conductivity can be treated successfully by the percolation theory [26] using the coverage fraction in two dimensions and the volume concentration in three-dimensional composites as the variable parameters. Resistivity in two-dimensional fractal topology can be described [26,27,28] as
(1)$$\rho \left(x\right)=\rho_{0}{\left(x-x_{c}\right)}^{-m}\;$$
where $\rho_{0}$ is the resistivity of the bulk solid with the same structure and composition as the film, exponent m is 1.2 for 2-dimensional systems, x is a fraction of the surface covered by the metal, and $x_{c}$ is the critical coverage fraction below which metallic clusters become finite. The coverage fraction is assumed to be proportional to the average thickness d of the deposited material, x∝d, therefore
(2)$$\rho \left(d\right)=\alpha \rho_{0}{\left(d-d_{c}\right)}^{-m}\;$$
where $d_{c}$ is the thickness percolation threshold and α—the units conversion coefficient. The percolation model predicts the divergence of resistivity at the finite thickness $d_{c}$.

We extend the model to the case in which a number of independent scattering mechanisms contribute to resistivity, such as the phonon, magnon, and lattice disorder scattering. The total bulk resistivity is given by Matthiessen’s rule as
(3)$$\rho_{0}=\sum \rho_{0i}$$
where $\rho_{0i}$ is the bulk resistivity component due to a particular scattering mechanism. If the conducting network in the fractal range is massive enough, such that each of these scattering mechanisms is preserved on a microscopic scale, then in the fractal range:(4)$$\rho_{i}\left(d\right)=\alpha \rho_{0i}{\left(d-d_{c}\right)}^{-m}$$

In other words, each resistivity component diverges at the percolation threshold separately. The same arguments apply when modification of the band structure adds a new scattering term to the bulk resistivity, such that
(5)$$\rho_{0}^{*}=\rho_{0}+\Delta \rho_{0}$$
where $\rho_{0}$ and $\rho_{0}^{*}$ are the total bulk resistivity of the original and modified states respectively, and $\Delta \rho_{0}$ is a bulk value of the additional scattering resistivity term. Following Equations (3) and (4), all terms diverge in the fractal range as
(6)$$\Delta \rho \left(d\right)=\alpha \Delta \rho_{0}{\left(d-d_{c}\right)}^{-m}\;\rho \left(d\right)=\alpha \rho_{0}{\left(d-d_{c}\right)}^{-m}\;\rho^{*}\left(d\right)=\alpha \rho_{0}^{*}{\left(d-d_{c}\right)}^{-m}$$

The normalized thickness-dependent extra-resistivity term is given by
(7)$$\frac{\Delta \rho \left(d\right)}{\rho \left(d\right)}=\frac{\alpha \Delta \rho_{0}{\left(d-d_{c}\right)}^{-m}}{\alpha \rho_{0}{\left(d-d_{c}\right)}^{-m}}=\frac{\Delta \rho_{0}}{\rho_{0}}$$

It is constant and equal to its bulk value.

The classical model assumes the resistivity of a uniform media at the onset of the fractal range to be that of the bulk $\rho_{0}$. However, it can be larger than the bulk value due to the surface and grain boundary scattering in thin films [29,30]. The value of the extra-scattering at the fractal range onset can also differ from the bulk. Therefore, we replace the bulk parameters $\Delta \rho_{0}$ and $\rho_{0}$ by their respective values $\Delta \rho_{0f}$ and $\rho_{0f}$ at the onset of the fractal range. The thickness-dependent extra-scattering resistivity component is then
(8)$$\Delta \rho \left(d\right)=\frac{\Delta \rho_{0f}}{\rho_{0f}}\rho \left(d\right)\;$$

It is expected to grow linearly with resistivity in the fractal range (Equation (8)) and diverge at the percolation threshold (Equation (6)). The resistivity of metallic conductometric sensors, such as gas sensors, increases due to additional scattering proportional to the amount of the absorbed gas. Their response Δρ(d) is expected to be given by Equations (6) and (8).

## 3. The Experiment

Samples used in this study were 2 nm to 100 nm thick polycrystalline Pd and Co_26_Pd_74_ alloy films with lateral dimensions 5 × 5 mm grown by radio frequency (RF)–magnetron sputtering onto room temperature glass substrates. Binary Co_26_Pd_74_ films were co-sputtered from separate targets (1.3″ diameter and 2 mm thick). Co and Pd are soluble and form an equilibrium fcc solid solution phase at all compositions during the room temperature deposition. No post-deposition annealing was carried out. The desired composition and thickness were controlled by the relative sputtering rates in the range 0.01–0.1 nm/sec with the respective RF power between 0 and 85 W and tested by EDS (energy-dispersive X-ray spectroscopy) measurements. Resistance was measured using the Van der Pauw protocol. Electrical contacts were attached by bonding Al/Si wires. An image of a wired sample within the sample holder is shown in Figure 1. The setup was equipped with a gas-control chamber, which enabled performing measurements at variable hydrogen concentrations. More details on the experimental setup can be found in Ref. [31]. The 15 nm thick and thinner Pd films were below the delamination thickness threshold [32] and were stable under repeated hydrogenation–dehydrogenation cycles. No buckling was observed in the CoPd samples at all tested thicknesses. The hydrogen-induced resistance changes were extracted from measurements performed in dry nitrogen and in 4% H_2_/N_2_ mixture gas at ambient pressure and temperature. The topology of the selected thin samples was studied by the transmission electron microscopy (TEM) using 5 nm thick films grown directly on transmission electron microscope grids.

## 4. Results and Discussion

A TEM micrograph of the 5 nm thick Pd film is shown in Figure 2. Polycrystalline fcc Pd (dark in the figure) with crystalline dimensions of 3–5 nm forms a typical percolation pattern. Films thicker than 15 nm are uniform. Figure 3 presents the resistivity of Pd films as a function of thickness. Two ranges can be identified. The resistivity of films with d ≥ 10 nm increases gradually with thickness reduction due to an enhanced surface scattering and can be well-fitted by the regular Fuchs–Sondheimer model [29,30]. A sharp resistivity growth starts below 6 nm. The main frame is a semi-logarithmic plot of resistivity as a function of thickness for samples below 6 nm thick. Open and solid circles present the resistivity prior to and after hydrogen loading, respectively. The solid lines are fits to Equation (2): $\rho \left(d\right)\propto {\left(d-d_{c}\right)}^{-m}$ for the hydrogen-free and hydrogenated states, respectively. The fits are good, which confirms that films have fractal topology, and their resistivity is well described by the percolation model. The power index m=1.1±0.1 and the critical thickness $d_{c}$ is 2.4±0.05 nm in both states, indicating that the surface coverage does not change during hydrogenation.

Absorption of hydrogen is expected to add the H scattering and enhance the material resistivity. However, as seen in Figure 3, the final resistivity of the hydrogenated samples is lower than the hydrogen-free ones. Two mechanisms have been suggested to explain the reduction in the resistivity instead of the anticipated enhancement: closure of intergranular gaps due to lateral expansion of inflating Pd clusters [33,34,35] and, alternatively, the out-of-plane thickness expansion [36]. The latter argued that thin films grown on rigid substrates cannot expand laterally within the film plane due to adhesion to the surface. Suppression of the in-plane expansion is equivalent to the application of the in-plane compressive stress leading to the out-of-plane thickness expansion and reduction in resistivity. The thickness of the hydrogenated film increases from $t_{0}$ before the hydrogenation to $t_{1}=\left(1+\gamma \right)t_{0}$ after the hydrogenation, where γ is the thickness expansion coefficient, which depends on the elastic parameters of the given material and the absorbed hydrogen concentration. The overall change in resistivity on hydrogen loading is given by
(9)$$\Delta \rho =\frac{\Delta \rho_{H}-\gamma \rho_{0}}{1+\gamma }$$
where $\rho_{0}$ is the initial resistivity and $\Delta \rho_{H}$ is the resistivity scattering term due to the absorption of hydrogen and formation of the hydride. The thickness expansion term ($-\gamma \rho_{0}$) depends on the initial resistivity, and is dominant in high resistivity thin films. A sign of the resistivity response to hydrogen was shown to change from positive in thick low resistivity films to negative in thin high resistivity films at the cross-over of about 40–60 μΩcm in a 4% H_2_/N_2_ atmosphere [36]. A positive H scattering and a negative thickness expansion terms can be identified and distinguished from each other by hydrogenation–dehydrogenation cycling [37]. The scattering mechanism is reversible when hydrogen is removed, while the thickness expansion is plastic and irreversible.

Figure 4 presents the resistivity response of 15 nm and 3 nm thick samples to a sequence of hydrogenation and dehydrogenation cycles (sequential exposure to 1 atm 4% H_2_/N_2_ gaseous mixture followed by N_2_). The entire sequence is a composition of reproducible rapid increase/decrease responses to the loading/unloading of hydrogen superposed with an irreversible gradual reduction in resistivity. The sequence can be interpreted as a superposition of reversible hydride formation–removal signals on the background of the irreversible thickness inflation [37]. Thickness expansion is much slower than hydrogen diffusion in and out of the film, and the process terminates after a number of cycles or, equivalently, after continuous hydrogenation for a period of 1–2 h. The rapid increase in resistance on hydrogen loading and the respective drop on hydrogen removal are due to the H scattering, which is the object of this study. Spikes in 15 nm thick sample are due to the simultaneous drop in H scattering with hydrogen removal and a minor increase in resistivity due to a partial thickness recovery.

The hydride scattering resistivity term $\Delta \rho_{H}$ is plotted in Figure 5 as a function of resistivity for films of different thicknesses. $\Delta \rho_{H}$ is about 6 μΩcm in thick films, decreases to 3 μΩcm in 10 nm thick film at the onset of the fractal range, and grows linearly with resistivity in thinner films. The slope $\frac{\Delta \rho_{H}\left(d\right)}{\rho \left(d\right)}$ is about 0.05, very close to the value $\frac{\Delta \rho_{Hf}}{\rho_{0f}}$ in 10 nm thick film at the onset of the fractal range. The results are in full agreement with the prediction of Equation (8).

Enhancement of the absolute resistivity response in the fractal thickness range can be particularly useful when the response in bulk films is too small to be detected. An example is demonstrated in Figure 6 for two Co_26_Pd_74_ films 60 nm (a) and 3 nm (b) thick. Ferromagnetic Pd-based materials, such as Co-Pd alloys and multilayers can be used for a combined resistive and magnetic detection of hydrogen [27,38]. The H scattering resistivity term in thick alloy films is significantly smaller than in pure Pd and is hardly visible on the thickness expansion background. On the other hand, the effect is clearly detectable in the thin 3 nm film.

## 5. Conclusions

To summarize, we extended the classical percolation model to the case in which the total resistivity is a superposition of several scattering mechanisms. The contribution of each mechanism is expected to grow and diverge at the conductance percolation threshold together with the total resistivity. The model is relevant to multiple circumstances in which changes in bulk resistivity induced by gas absorption or modifications in the band structure by pressure, phase transitions, intercalation, etc. are compared with those observed in thin films of the same material in the fractal geometry range. We tested the model experimentally using the hydrogenated Pd and ferromagnetic CoPd alloy films where absorption of hydrogen induces an additional electron scattering by H atoms occupying the interstitial lattice sites. The hydrogen scattering resistivity term was found to grow linearly with the total resistivity and diverge at the percolation threshold in agreement with the model. Enhancement of the absolute value of the resistivity response in thin film sensors is particularly important when the respective bulk material response is too small for reliable detection.

## Figures and Tables

**Figure 1 sensors-23-02409-f001:**
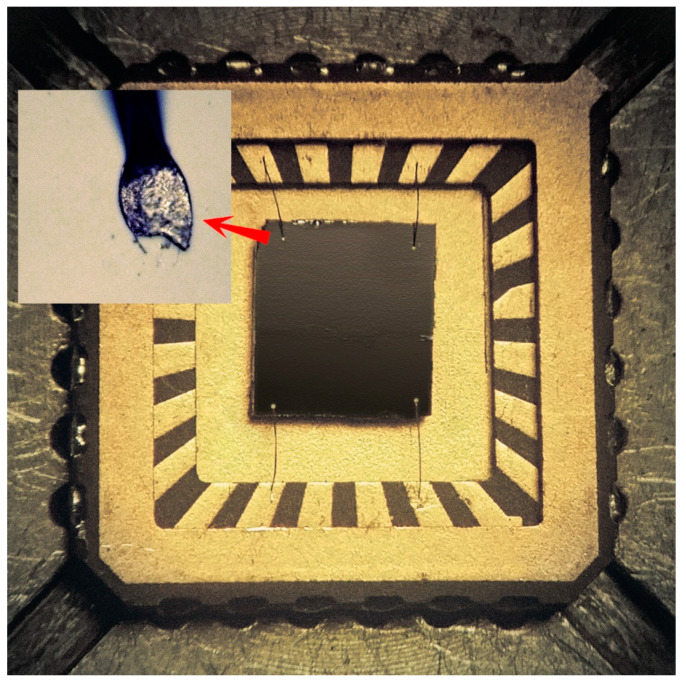
Image of 5 × 5 mm sample mounted in the sample holder. Al/Si wires are attached by bonding.

**Figure 2 sensors-23-02409-f002:**
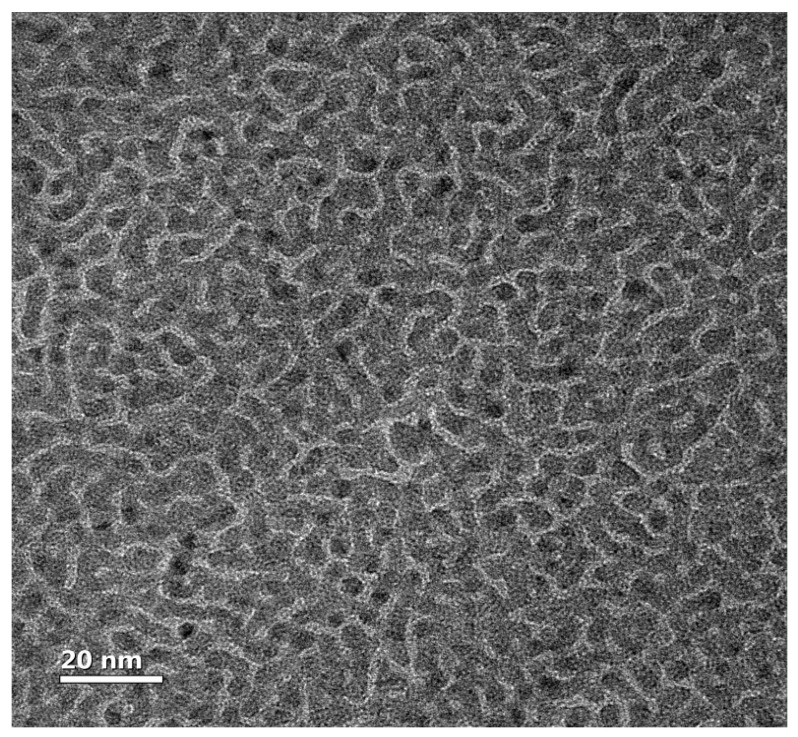
TEM micrograph of the 5 nm thick Pd film. Pd is dark.

**Figure 3 sensors-23-02409-f003:**
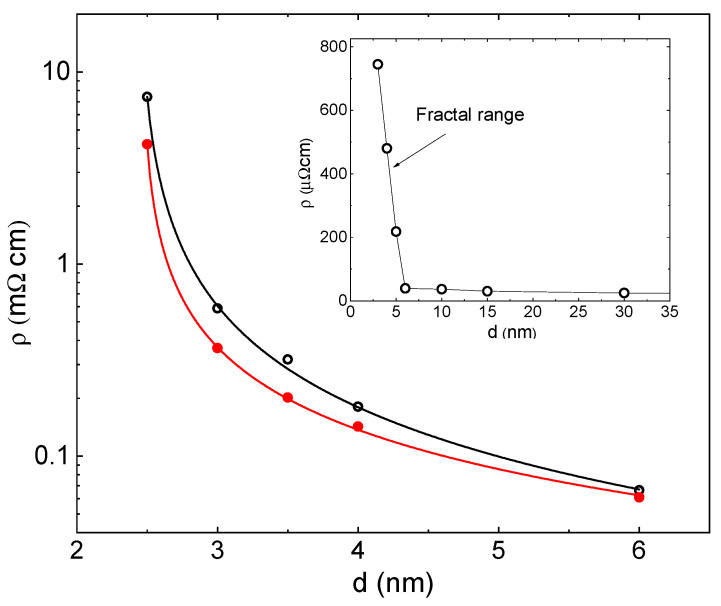
Resistivity of Pd films as a function of thickness prior (open circles) and after (solid circles) hydrogen loading. Solid lines in the main frame fit to Equation (2): $\rho \left(d\right)=\alpha \rho_{0}{\left(d-d_{c}\right)}^{-m}$. $d_{c}$ = 2.4±0.05 nm and m=1.1±0.1 in both states. Inset: resistivity as a function of thickness prior to hydrogen loading in a linear scale.

**Figure 4 sensors-23-02409-f004:**
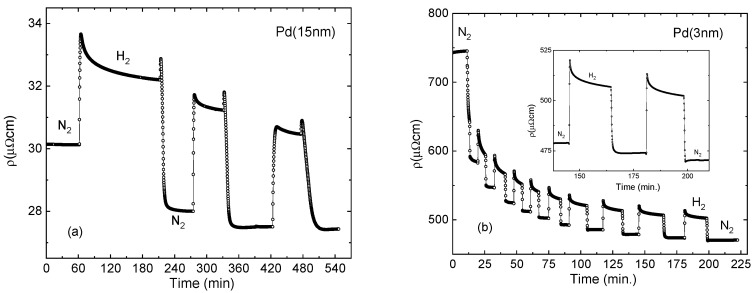
Resistivity response of 15 nm (**a**) and 3 nm (**b**) thick Pd samples to the sequences of hydrogenation and dehydrogenation cycles (sequential exposure to 1 atm 4% H_2_/N_2_ gaseous mixture followed by N_2_). The sharp increase in resistance on hydrogen loading and the respective drop on hydrogen removal are due to the hydrogen scattering.

**Figure 5 sensors-23-02409-f005:**
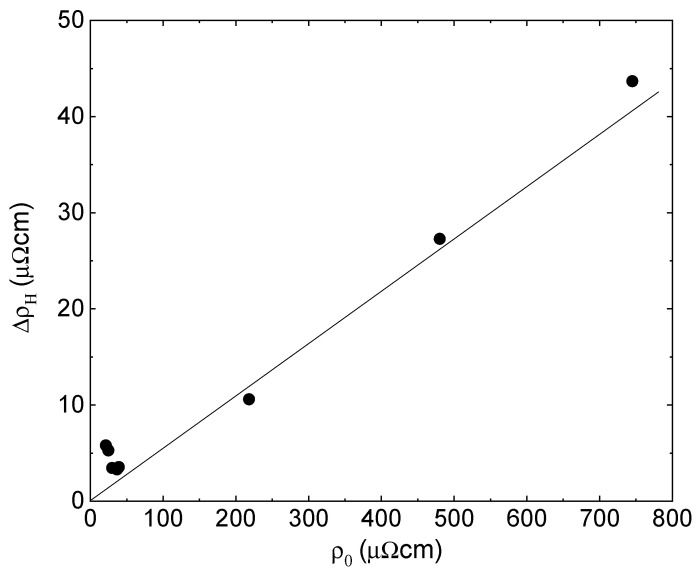
The hydride scattering resistivity term $\Delta \rho_{H}$ as a function of resistivity for films of different thicknesses. A solid line is a guide to the eye.

**Figure 6 sensors-23-02409-f006:**
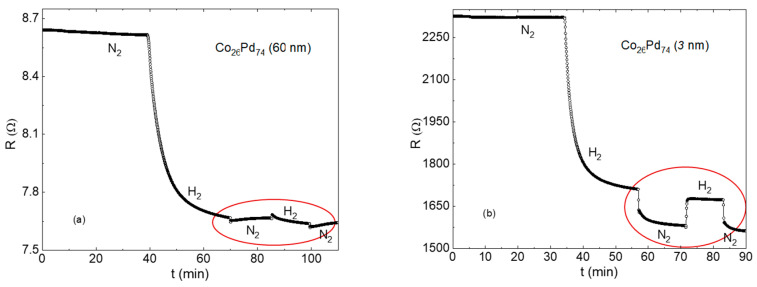
Resistance response of 60 nm (**a**) and 3 nm (**b**) thick Co_26_Pd_74_ films to the sequences of hydrogenation and dehydrogenation cycles. The H scattering terms are marked by ellipses. The background resistance decrease is due to a gradual thickness expansion in the hydrogen atmosphere.

## Data Availability

The data that support the findings of this study are available within the article.

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
