# Peer review of "Effect of Fractal Topology on the Resistivity Response of Thin Film Sensors"

_sensors, 2023, doi:10.3390/s23052409_

Round 1

Reviewer 1 Report

The authors studied thin metallic conductometric sensors with large resistivity response to external stimuli. They proposed a percolation model and tested it experimentally using thin films of hydrogenated palladium and CoPd alloys, and found that absorbed hydrogen atoms occupy the interstitial lattice sites and enhance the electron scattering. The hydrogen scattering resistivity was found to grow linearly with the total resistivity in the fractal topology range. The manuscript is well written and can be accepted before minor revision.
1. Fig. 2 shows resistivity of Pd films as a function of thickness before and after hydrogen loading. Could you also provide the details of fractal topology as given in the title of this manuscript.
2. Please also provide more details about the mechanism of thickness effect.

Author Response

  1. Fig. 2 shows resistivity of Pd films as a function of thickness before and after hydrogen loading. Could you also provide the details of fractal topology as given in the title of this manuscript.

We expanded the description of the fractal topology and added the relevant references on page 2.

  1. Please also provide more details about the mechanism of thickness effect.

More details and Eq.9 were added on page 7

Reviewer 2 Report

1. The introduction section regarding thin-film sensors needs improvement. phrases such as: "However, in the very thin limit, the material is not homogeneous covering only a fraction of the surface" need backing up with references, as nanometer-thin sensitive films were proven, in the literature for the past decade, to be even across the sensor surface, if an appropriate deposition technique is used (sol-gel, screen-printing, vapor techniques CVD or PVD).

2. A schematic representation of the used sensor prototype with film deposition and type of electrodes and appropriate sizing is always required for sensing related articles. Please insert one separate figure containing the sensor prototype in the experimental section.

4. No mentions on the types of electrodes used in the sensor transducer or regarding the sensing cell. Is it a commercially available transducer type utilized for fabricating this sensor? Please make a more comprehensive description in the manuscript regarding the gas-sensing experimental setup and clarify those aspects.

3. Newly used abbreviated terms need defining for the reader, e.g: "grown by rf-magnetron sputtering". Then "rf" is used again in the manuscript as is.

Author Response

  1. The introduction section regarding thin-film sensors needs improvement. phrases such as: "However, in the very thin limit, the material is not homogeneous covering only a fraction of the surface" need backing up with references, as nanometer-thin sensitive films were proven, in the literature for the past decade, to be even across the sensor surface, if an appropriate deposition technique is used (sol-gel, screen-printing, vapor techniques CVD or PVD).

A description of the early stages of the thin film growth and the relevant references were added in the introduction section on page 2.

  1. A schematic representation of the used sensor prototype with film deposition and type of electrodes and appropriate sizing is always required for sensing related articles. Please insert one separate figure containing the sensor prototype in the experimental section.

Figure 1 illustrating the wired sample within the sample holder was added. The sample dimensions are specified in the Experimental section (Samples used in this study were 2 nm to 100 nm thick polycrystalline Pd and Co­26Pd74 alloy films with lateral dimensions 5 x 5 mm grown by radio frequency (RF)-magnetron sputtering onto room-temperature glass substrates).

  1. No mentions on the types of electrodes used in the sensor transducer or regarding the sensing cell. Is it a commercially available transducer type utilized for fabricating this sensor? Please make a more comprehensive description in the manuscript regarding the gas-sensing experimental setup and clarify those aspects.

The type of electrodes is mentioned in the Experimental section (Electrical contacts were attached by bonding Al/Si wires) and illustrated in the added figure 1.

  1. Newly used abbreviated terms need defining for the reader, e.g: "grown by rf-magnetron sputtering". Then "rf" is used again in the manuscript as is.

Corrected.

Round 2

Reviewer 2 Report

All comments were addressed. Manuscript was improved.